# Uncovering specific mechanisms across cell types in dynamical models

**Adrian L. Hauber**[1,2,3]*, **Marcus Rosenblatt**[1,2], **Jens Timmer**[1,2,3]

**1** Institute of Physics, University of Freiburg, Freiburg, Germany, **2** Freiburg Center for Data Analysis and Modeling, University of Freiburg, Freiburg, Germany, **3** Centre for Integrative Biological Signalling Studies, University of Freiburg, Freiburg, Germany

* adrian.hauber@fdm.uni-freiburg.de

**Data Availability Statement:** The method of regularization with symmetric penalization of fold-change parameters is implemented in the open-source Matlab toolbox Data2Dynamics and is accessible via GitHub https://github.com/

## Abstract

Ordinary differential equations are frequently employed for mathematical modeling of biological systems. The identification of mechanisms that are specific to certain cell types is crucial for building useful models and to gain insights into the underlying biological processes. Regularization techniques have been proposed and applied to identify mechanisms specific to two cell types, e.g., healthy and cancer cells, including the LASSO (least absolute shrinkage and selection operator). However, when analyzing more than two cell types, these approaches are not consistent, and require the selection of a reference cell type, which can affect the results. To make the regularization approach applicable to identifying cell-type specific mechanisms in any number of cell types, we propose to incorporate the clustered LASSO into the framework of ordinary differential equation modeling by penalizing the pairwise differences of the logarithmized fold-change parameters encoding a specific mechanism in different cell types. The symmetry introduced by this approach renders the results independent of the reference cell type. We discuss the necessary adaptations of state-of-the-art numerical optimization techniques and the process of model selection for this method. We assess the performance with realistic biological models and synthetic data, and demonstrate that it outperforms existing approaches. Finally, we also exemplify its application to published biological models including experimental data, and link the results to independent biological measurements.

## Author summary

Mathematical models enable insights into biological systems beyond what is possible in the wet lab alone. However, constructing useful models can be challenging, since they both need a certain amount of complexity to adequately describe real-world observations, and simultaneously enough simplicity to enable understanding of these observations and precise predictions. Regularization techniques were suggested to tackle this challenge, especially when building models that describe two different types of cells, such as healthy and cancer cells. Typically, both cell types have a large portion of biological mechanisms in common, and the task is to identify the relevant differences that need to be included into the model.

Data2Dynamics/d2d. A demonstration script can be found under arFramework3/Examples/ Becker_Science_2010/Setup_Regularization.m.

**Funding:** This work was funded by the German Research Foundation (DFG) under Germany's Excellence Strategy (CIBSS – EXC-2189 – Project ID 390929984; A.H.), the SFB 1381 (Project ID 403222702, A.H.), and the TRR 179 (Project ID 272983813, M.R.). We acknowledge support by the state of Baden-Württemberg through bwHPC and the German Research Foundation through grant INST 35/1134-1 FUGG. The funders had no role in study design, data collection and analysis, decision to publish, or preparation of the manuscript.

**Competing interests:** The authors have declared that no competing interests exist.

For more than two types of cells, the existing approaches are not readily applicable, because they require defining one of the cell types as reference, which potentially influences the results. In this work, we present a regularization method that is independent from the choice of a reference. We demonstrate its working principle and compare its performance to existing approaches. Since we implemented this method in a freely available software package, it is accessible to a broad range of researchers and will facilitate the construction of useful mathematical models for multiple types of cells.

## I. Introduction

Mechanistic modeling by means of ordinary differential equations (ODEs) has become a wide-spread method to understand and discover systemic behavior and dynamic information processing of complex biological systems. Along with the development of experimental techniques such as quantitative high-throughput measurements, these mathematical models tend to become more and more complex with hundreds of parameters that have to be calibrated to match experimental data. This manifests in (i) more elaborate biological questions that can only be addressed by taking into account more biological components and corresponding mechanisms [1–3], (ii) accounting for broad ranges of input doses and time scales to enlighten the full dynamical information [4–7], but also (iii) the need of such models to be valid across multiple biological systems, e.g., different cell types, model organisms, or patients [8], which roughly multiplies the number of involved model parameters by the number of systems.

In the setting of $n$ cell types, a typical assumption is that their dynamics can be described by ODE systems with an identical structure but potentially different parameter values, e.g., accounting for mutations [9] or copy number variations [10] between different cell types. The challenge is then to cluster the cell types into groups that share the same value for a certain parameter which reduces the overall number of parameters. From the opposite perspective, the task of balancing model complexity for ODE models of $n$ related biological systems, e.g., cell types, can also be understood as the task of determining which of the involved parameters need to be specific to the cell type to, both, explain experimental observations and keeping model complexity as low as possible. This parameter selection is related to the general topic of feature selection or model discrimination [11], and was recently solved for the case $n = 2$: [12] transferred the concept of the LASSO (least absolute shrinkage and selection operator) for regression [13] to the field of parameter estimation in ODE models and employed an optimization strategy outlined in [14]. They adapted *Matlab's* trust-region optimizer *lsqnonlin* to be capable of handling the discontinuous derivative occurring in the involved $L_1$-norm at zero. Following on this study, [15] compared different regularization terms that include sparsity in the system with the interesting result of the $L_{0.8}$-penalization outperforming both the classical $L_1$ approach and the elastic net, i.e., a regularization with a combination of $L_1$ and $L_2$ norms, with respect to reliability of detecting sparsity and optimization performance.

For the case of $n > 2$ cell types, the methodology of [12] and [15] can also be applied. Since the regularization term is no longer symmetric with respect to changing the labeling of the cell types, the choice of the reference cell type potentially influences the result of the regularization. To resolve this asymmetry, we propose to employ a regularization function that includes penalty terms for differences in fold change parameters between all $n(n−1)/2$ possible pairs of cell types. This corresponds to the regularization applied in the clustered LASSO that was proposed for regression modeling [16]. In the context of mechanistic ODE models, it allows, e.g., clustering of cell types into groups that share identical parameters, and thereby enables the discovery of any kind of sparsity structure in the set of parameters. When sensible groups of

**Table 1. Examples of applicability of regularization methods for different parameter subgroup structures.**

| Description | Cell Type 1 (Reference) | Cell Type 2 | Cell Type 3 | Suitable Method |
|---|---|---|---|---|
| Subgroup including reference | 1 | 1 | 2 | LASSO, Clustered LASSO |
| No subgroups | 1 | 2 | 3 | Group LASSO, Clustered LASSO |
| Subgroup excluding reference | 1 | 2 | 2 | Clustered LASSO |

parameters can be predefined by including prior knowledge, the grouped LASSO [17] can be applied. However, in the setting of different healthy or cancer cell-lines, which typically include recurrent mutations [18], the flexible identification of sparsity enabled by our approach is essential. Also, it is not necessary to define parameter groups beforehand. An overview of the applicability of the different methods is provided in Table 1.

Within this publication, we provide a systematic approach to detect and quantify cell type-specific parameters and thereby enable a statistically sound reduction of the remaining non-specific parameters in ODE models. The main goal of inferring cell type-specific and non-specific parameters is to identify mechanisms that are different between related biological systems and those that are shared across them. We demonstrate the necessary adaptions to the optimization algorithm and model selection to incorporate the symmetric regularization function into the framework of ODE modeling. Finally, we provide an assessment of the performance of the method we propose, and apply it to biological data.

## II. Problem statement

### Parameter estimation in dynamical systems

Biochemical pathway models can be formulated as dynamical systems by means of ordinary differential equations (ODEs) which are based on *a priori* knowledge about underlying mechanisms:

$$\dot{\vec{x}}(t) = \vec{f}(\vec{x}(t), u(t), \vec{p})$$

Such models typically comprise unknown but constant parameters $\vec{p}$, which represent, e.g., reaction rate constants, or initial conditions of the dynamical system. The biochemical species are contained in the state vector $\vec{x}$. Experimental perturbations are incorporated through the input $u(t)$. Maximum likelihood estimation combined with high performance numerical optimization methods provides a statistically sound and efficient way to infer the values of parameters from data $\{y^*_{i,t}\}$ with normally distributed errors $\epsilon_{i,t} \sim N(0, \sigma_{i,t}^2)$. The model states are linked to predictions of experiments $y_i$ via the observation function:

$$y_i(t, \vec{p}) = g_i(\vec{x}(t, \vec{p}), \vec{p}) + \epsilon_{i,t}$$

Model calibration is equivalent to minimization of

$$\chi^2(\vec{p}) := \sum_{i,t} \frac{[y^*_{i,t} - y_i(t, \vec{p})]^2}{2\sigma_{i,t}^2}, \tag{1}$$

where all parameters are usually estimated on the logarithmic scale, rendering them strictly positive.

### Multiple cell types as related biological systems

A common application for regularization in systems biology is the identification of cell-type specific parameters in ODE models [9,12]. In practice, such a model is constructed and

calibrated with the data of only one cell type initially, which we will regard to as the reference cell type. Typically, one assumes that the other cell types share at least the model structure with the reference cell type, if not exhibit identical behavior. Therefore, the starting point for modeling of the other cell types are copies of the model for the reference cell type, which, however, are allowed to comprise different parameter values for the different cell types. When *a priori* knowledge on common parameter values, e.g., the time scale of a protein degradation is available, it is also possible to allow only a subset of parameters to be specific to the cell types [19].

Let $p_i^{(j)}, j = 1, 2, \ldots, n$ denote mechanistically equivalent parameters in $n$ models for $n$ different cell types. For example, $p_i^{(j)}$ might have one specific value in wild-type cells, and a different value in mutated cells. Throughout this paper, we will denote the reference cell type with $j = 1$. Consequently, the cell type-specific parameters read $p_i^{(j)} = p_i^{(1)} \tilde{r}_i^{(j)}$, where $\tilde{r}_i^{(j)}$ represents the fold change that relates cell type $j$ to the reference for parameter $i$, and $\tilde{r}_i^{(1)} = 1$. On the logarithmic scale, the transformation that relates the cell type-specific parameters to those of the reference cell type reads

$$\log p_i^{(j)} = \log p_i^{(1)} + r_i^{(j)}, r_i^{(j)} = \log \tilde{r}_i^{(j)}. \tag{2}$$

## LASSO regularization for the case of *n* = 2 cell types

In systems biology, a common application of regularization is the identification of shared mechanisms or mutations in a biochemical reaction network across multiple cell types. The term *regularization* refers to amending the objective function $L$ by an additional function $v(\overrightarrow{r})$:

$$L(\overrightarrow{p}, \overrightarrow{r}, \lambda) = \chi^2(\overrightarrow{p}, \overrightarrow{r}) + \lambda v(\overrightarrow{r}), \tag{3}$$

where $\lambda$ is the regularization strength, which is *a priori* an unknown constant. To find common mechanisms across multiple cell types, LASSO regularization can be applied with penalization of deviations from zero of the $L_1$ norm of the logarithmized fold change parameters (Fig 1A; [12]), i.e.,

$$v(\overrightarrow{r}) = \sum_{i,j} |r_i^{(j)}|, \tag{4}$$

which has proven to be useful in the context of (logical) ODE models [9,20–24]. Further, it needs to be emphasized that in the context of ODE models with log-transformed parameters, the $L_1$ norm has been reported as suboptimal due to the alignment of equal penalization manifolds with manifolds of equal likelihood induced by the model structure, and employing an $L_q$ pseudonorm with $q = 0.8$ was suggested [15].

## The challenge of *n>2* cell types

When $n>1$ cell types are analyzed, the problem is not symmetric w.r.t. the choice of the reference cell type anymore, because by means of Eq (4), only deviations from the reference are penalized. Deviations between pairs of the $n−1$ non-reference cell types remain unaffected by the regularization, i.e., shared mechanisms between cell types that do not include the reference cannot be detected. Therefore, the result of regularization potentially depends on the *a priori* choice of the reference cell type.

Additionally, the objective function value at a given point in parameter space depends on the choice of the reference cell type. Consider the simple example of three cell types differing in their values of $p$ at a point in parameter space that corresponds to log $p$ taking the values 0.5, 0.75, and 1, in the cell type 1,2, and 3, respectively. Using cell type 1 or 3 as a reference, the contribution to the objective function at this point is $|0.25|^q + |0.5|^q$ (Eq 4), while for cell type 2 as a

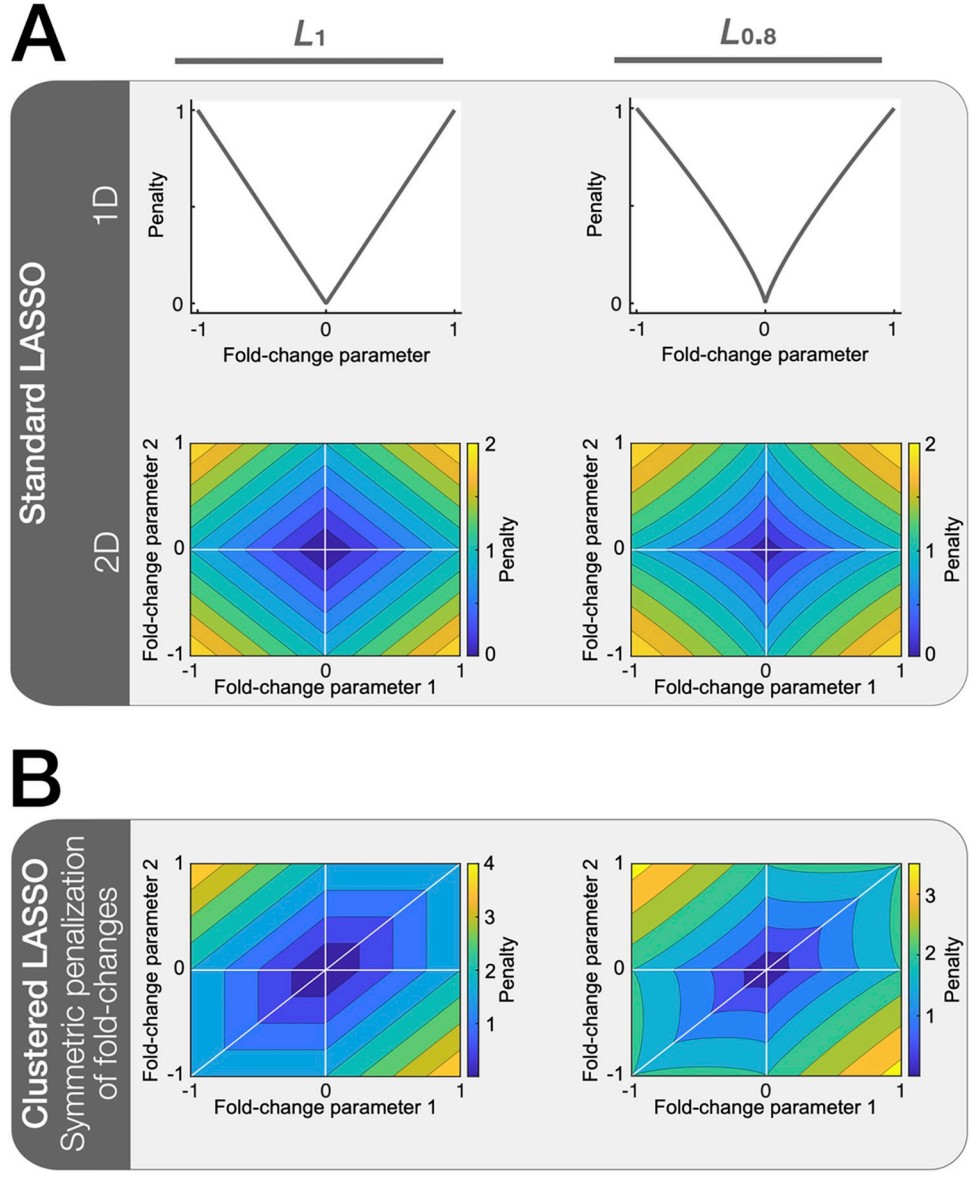

**Fig 1. (A)** Plots of the regularization function for the standard LASSO approach, i.e., penalization of logarithmized fold-change parameter values different from zero, and penalization landscapes for the two-dimensional case. **(B)** Regularization function for the symmetric penalization of fold-change differences via the clustered LASSO, evaluated each for the $L_1$ and $L_{0.8}$ penalizations.

reference, it is only $2|0.25|^q$. Since the regularization is stronger in the former case, it is also more likely to alter the position of the optimum induced by the data contribution (Eq 3). Therefore, the endpoint of regularized optimization and also the resulting clusters potentially depend on the choice of the reference cell type.

## III. Methods

### Symmetric penalization of fold-changes

The regularization function in Eq (4) induces a dependence on the choice of the reference cell type, because it only penalizes fold-changes that represent pairwise differences between each

cell type and the reference cell type. Therefore, we propose to employ the clustered LASSO regularization regularization function

$$v(\overrightarrow{r}) = \sum_{i,j<k} |r_i^{(j)} - r_i^{(k)}|^q, \tag{5}$$

taken from [16] and adapted to the setting of differential equation modeling. In this way also the pairwise differences are penalized and symmetry is reassured. In some sense, this can be interpreted as iterating over all possible reference cell types and summing up the penalty terms. In addition, the chosen regularization function (Eq 5) employs $L_q$ penalization to facilitate optimization in the setting of ODE models with log-transformed parameters (Fig 1B). Note, that the penalization introduced by Eq (4) is included here through the terms with $j$, $k = 1$, since $\tilde{r}_i^{(1)} = 1$, i.e., $\log r_i^{(1)} = 0$ and $q = 1$. With the proposed regularization function, clusters of any size within the $n$ cell types that share the same parameter value are promoted, implying that they for example exhibit the same mutation or are governed by the same mechanism.

We propose to use the $L_{0.8}$ pseudonorm also with the clustered LASSO regularization function, since firstly it was found to be a suitable heuristic for the choice of $q$ in the context of standard LASSO regularization [15], and secondly our analysis indicates a superior performance compared to $q = 1$ (Section IV). In contrast to [16], we employ the same regularization strength for all penalization terms corresponding to one parameter $p_i$, because the problem is completely symmetric w.r.t. the choice of the reference cell-line, and hence, penalization of fold-change parameters and differences of such must be treated equally.

Due to this symmetry, the choice of the reference cell-line has no effect on the outcome of the regularization procedure. Because defining a reference cell-line is still beneficial for technical reasons, it will be denoted as the *technical reference* in the following. For general limitations of the clustered LASSO, see [16].

## Adaptations to the optimization algorithm

Optimization algorithms, such as *Matlab*'s *lsqnonlin*, frequently require the objective function $L$ to be formulated as a sum of squared residuals. In terms of data contribution (Eq 1), this reads $L(\overrightarrow{p}, \overrightarrow{r}, \lambda) = \sum_i \text{res}_i^2$. In our case of clustered LASSO regularization (Eq 5), an additional $\binom{n}{2} = n(n-1)/2$ residuals need to be considered, analogously to the case of the standard LASSO [12]:

$$\text{res}_m = \sqrt{\lambda |r_i^{(j)} - r_i^{(k)}|^q}$$

The so-called sensitivities $\text{sres}_{lm} = \partial \text{res}_m / \partial p_l$ represent the rate of change of a residual w.r.t. a certain parameter, which is a necessary value to be handed over to the optimizer and is employed during each step of the optimization. Let $r_i^{(j)}$ be the $l$-th and $r_i^{(k)}$ be the $n$-th parameter of the model. The sensitivities associated with the regularization residual $\text{res}_m$ then read:

$$\text{sres}_{lm} = -\text{sres}_{om} = \frac{q}{2} \sqrt{\lambda |r_i^{(j)} - r_i^{(k)}|^{q-2}} \text{sgn}\left(r_i^{(j)} - r_i^{(k)}\right) \quad \text{for } r_i^{(j)} - r_i^{(k)} \neq 0$$

**Optimality criterion in presence of regularized fold-change differences.**   Without regularization, the process of optimization can be considered complete when $\overrightarrow{\nabla} \chi^2(\overrightarrow{p}) = 0$, i.e., the objective function landscape induced by the data has no slope at the current parameter vector

$\vec{p}$. When optimizing with the regularization term for symmetric penalization of fold-changes (Eq 5), this criterion on the level of the individual parameters must be extended to

$$\vec{\nabla}_{\vec{p}}\chi^2(\vec{p}, \vec{r}) = 0 \ (6.1) \text{ and either}$$

$$[\vec{\nabla}_{\vec{r}}L(\vec{p}, \vec{r})]_{l-n} = 0 \ \text{ for } |r_i^{(j)} - r_i^{(k)}| > 0 \ (6.2) \text{ or}$$

$$|[\vec{\nabla}_{\vec{r}}\chi^2(\vec{p}, \vec{r})]_{l-n}| < \lambda |[\vec{\nabla}v(\vec{r})]_{l-n}| \ \text{ for } |r_i^{(j)} - r_i^{(k)}| = 0 (6.3),$$

where the subscript l-n after denotes the gradient in the direction of $\vec{e_l} - \vec{e_n}$, and $\vec{e_n}$ being the unit vector to the n-th parameter axis.

Criterion 6.1 is the extension of the optimality criterion without regularization to an objective function that also depends on the fold-change parameters $\vec{r}$, for which optimality is determined by one of the following two criteria: Either the slope of the objective function induced by the data in the direction of $r_i^{(j)} - r_i^{(k)}$, and the slope of the regularization function in that direction exactly compensate each other at one point in parameter space, which is represented by criterion 6.2, or, the regularization function outweighs the data contribution in a point where two fold-change parameters are equal (Criterion 6.3).

The gradient of the regularization function (Eq 5) can be calculated:

$$[\vec{\nabla}v(\vec{r})]_l = q|r_i^{(j)} - r_i^{(k)}|^{q-1}\text{sgn}(r_i^{(j)} - r_i^{(k)}) \text{ for } r_i^{(j)} - r_i^{(k)} \neq 0.$$

For $r_i^{(j)} - r_i^{(k)} = 0$, the above expression diverges, which would introduce an optimum regardless of the data contribution $\chi^2(\vec{p}, \vec{r})$. To prevent optimization from getting stuck at this spurious optimum, gradients are evaluated not at this singularity, but at $\epsilon = 10^{-10}$ instead, and all $|r_i^{(j)} - r_i^{(k)}| < \epsilon$ are considered zero [15].

**Implementation of the optimality criterion.** The implementation of criterion 6.3 requires special attention: The termination of optimization due to arriving in a manifold where two fold-change parameters are equal and the regularization dominates the total objective function gradient (Criterion 6.3) can be implemented by manipulation of the sensitivity matrix such that the next optimization step does not change the value of $r_i^{(j)} - r_i^{(k)}$. For standard LASSO regularization, this can be ensured by setting all sensitivities corresponding to the involved fold-change parameter to zero [12]. However, for the symmetric penalization of fold-changes, this approach is not suitable, because it would terminate optimization prematurely: If at any point during optimization, the case $r_i^{(j)} - r_i^{(k)} = 0, r_i^{(j)} \neq 0$ occurs, the next step would not change the values of either $r_i^{(j)}$ or $r_i^{(k)}$, which is also true for all subsequent steps. Therefore, it would not be possible to ever reach the point $r_i^{(j)} = r_i^{(k)} = 0$, which should be the ultimate optimization endpoint for $\lambda \to \infty$.

We propose to employ an alternative method that ensures that optimization is terminated correctly in the setting of symmetric penalization of fold-changes. While it is a necessary condition to set the sensitivities corresponding to the regularization residual and $r_i^{(j)}$ and $r_i^{(k)}$ to the same value, this value does not have to be zero. Instead, we propose to use the mean of sensitivities corresponding to $r_i^{(j)}$ and $r_i^{(k)}$ for every residual, i.e.,

$$\text{sres}_{lm} \to \text{mean}(\text{sres}_{lm}, \text{sres}_{nm}) \ \forall m \text{ and}$$

$$\text{sres}_{nm} \to \text{mean}(\text{sres}_{lm}, \text{sres}_{nm}) \ \forall m,$$

which avoids zero-valued sensitivities wherever possible, and employs information from both individual sensitivities. We compared the performance to using the maximum of absolute values of sensitivities and found that using the mean resulted in better performance (S1 Text).

On the other hand, if at any point during optimization, criterion 6.3 is not fulfilled anymore, because the data contribution to the gradient indicates a step away from $|r_i^{(j)} - r_i^{(k)}| = 0$, the sensitivity corresponding to the respective regularization residual $\text{res}_{m^*}$ is set to zero, to allow the exploration of alternative optima:

$$\text{sres}_{lm^*} \rightarrow 0$$

$$\text{sres}_{nm^*} \rightarrow 0$$

**Optimization step truncation.** Due to the discontinuity in the derivative of the $L_q$ regularization terms at sign-changes of the fold-change parameters, optimization step truncation was suggested by [12] to enable efficient optimization. In the setting of symmetric penalization of fold-changes, we truncate optimization steps to prevent sign changes also in all $r_i^{(j)} - r_i^{(k)}$. If such a sign change would occur, we make a step directly to $r_i^{(j)} - r_i^{(k)} = 0$ instead, where the optimality criterion discussed above is evaluated before a subsequent step is performed.

## Selection of the parsimonious model

Regularized optimization promotes sparsity, because $r_i^{(j)} - r_i^{(k)} = 0$ minimizes the penalty induced by the regularization function (Eq 5) and corresponds to the effect encoded by parameter $p_i$ to be identical between cell type $j$ and $k$, which leads to a reduced number of degrees of freedom in the model. On the other hand, the data contribution to the objective function will almost over-fit finite amounts of data to maximize the goodness-of-fit and promote $r_i^{(j)} - r_i^{(k)} \neq 0$. Depending on the regularization strength $\lambda$, both the data and the regularization contribution to the objective function (Eq 3) determine the optimal value for $\lambda$. Finding a value for the regularization strength that balances model parsimony with goodness-of-fit is therefore crucial for using regularization to construct useful mathematical models.

A two-step model selection approach was proposed to identify the optimal regularization strength $\lambda^*$ [12]. First, optimization of the regularized objective function is performed for a discrete set of regularization strengths ranging usually over several orders of magnitude. With the model structure constrained to the clusters $r_i^{(j)} - r_i^{(k)} = 0$ identified in the first step, the objective function is then optimized a second time without regularization for each $\lambda$, to obtain unbiased parameter estimates. Finally, for each $\lambda$, a statistical test, e.g., a likelihood ratio test, is performed w.r.t the unregularized objective function $L(\overrightarrow{p}, \overrightarrow{r}, \lambda = 0)$. The optimal regularization strength is then given by the largest value for which the constrained model is not rejected to be consistent with the data by the statistical test.

To evaluate the statistical test, usually the number of degrees of freedom in the two alternative models must be taken into account. For the standard LASSO regularization, these are the number of fitted parameters minus the number of fold-change parameters that are equal to zero. For the symmetric penalization of fold-changes applied here, it is important to correctly take into account the case $r_i^{(j)} - r_i^{(k)} = 0$. This can occur for $r_i^{(j)} \neq 0$ and $r_i^{(k)} \neq 0$, which corresponds to reduction in the number of degrees of freedom by one, or alternatively, when $r_i^{(j)} = r_i^{(k)} = 0$. When counting the number of fold-changes and differences of fold-changes that are equal to zero, the latter case should reduce the number of degrees of freedom by two

only, i.e., the number of degrees of freedom is given by

$$m_{\lambda} = \#r_i^{(1)} - \#(r_i^{(1)} = 0)_{\lambda} - \#(r_i^{(j)} - r_i^{(k)} = 0 \& r_i^{(j)} \neq 0)_{\lambda}.$$

In the following, we use a likelihood ratio test with $\alpha = 0.05$, which follows the work of Steiert et al. The test statistic then reads

$$D(\lambda) = L(\overrightarrow{p}, \overrightarrow{r}, \lambda) - L(\overrightarrow{p}, \overrightarrow{r}, \lambda = 0),$$

which, according to Wilks' theorem [25], is distributed as a chi-squared distribution with $m_{\lambda}$ degrees of freedom.

## IV. Discussion

We implemented routines for the LASSO regularization with symmetric penalization of fold-changes into the open-source modeling environment for dynamical systems, Data2Dynamics [26]. It also includes routines for the standard LASSO regularization [12], to which we compare our method. Our implementation is applicable to all classes of models and data that can be implemented into Data2Dynamics.

### Application to a toy model with simulated data

**Model description.**   To showcase our method of symmetric penalization of fold-changes, we apply it to the following toy model that depicts the exponential decay of a species $x$ with the rate constant $p$ (Fig 2A):

$$\dot{x}(t) = -px(t).$$

For simplification, we assumed $x(t = 0) = 1$ for the initial concentration and a direct observation of the state $x$, i.e., $g(x(t),p) = x(t)$, such that $p$ is the only remaining free parameter. As a ground truth, we chose three cell types with a value for log p of -1.5 for cell type 1, -1.3 for cell type 2 and -1.2 for cell type 3 (Fig 2B), translating to the fold change parameters being $r^{(2)} = 0.2$, $r^{(3)} = 0.3$. We simulated data simulated with normally distributed experimental errors with log $\sigma_t = -1.3$ (Fig 2C). The parameter values were chosen such that $p$ has similar values in cell type 2 and 3, which are both clearly different from the value in cell type 1, and can be thought of as a common mutation in cell types 2 and 3. For illustrative purposes, $^{(2)}$ and $^{(3)}$ were chosen to differ slightly.

**Approach.**   Next, the toy model was fitted to the simulated data with the aim of retrieving the true parameter values stated above. We compared the application of three different regularization approaches: (i) Standard LASSO regularization, (ii) the proposed symmetric penalization of fold-changes with the $L_{0.8}$ pseudonorm, and (iii) with an $L_1$ norm instead. Each scenario was implemented with cell type 1 as the technical reference to be able to compare the results. For reasons of simplicity, we assume the true value of $p$ in the technical reference cell type is known, so that the resulting parameter space has only two dimensions.

For five values of $\lambda$, the objective function was shown in the $r^{(2)}$–$r^{(3)}$-plane together with the respective optimization endpoints after each step of increasing $\lambda$ (Fig 2D). Depending on the regularization strength $\lambda$, the total objective function changes from being dominated by the data through $\chi^2$, to mainly depicting the regularization function $v(\overrightarrow{r})$ (Eq 3). Note that in this visualization of the optimization path, each point on the diagonal $r^{(2)}$–$r^{(3)} = 0$ corresponds to a common mutation in cell types 2 and 3 as used for simulation, apart from the origin $r^{(2)} = r^{(3)} = 0$ where all three cell types would comprise the same parameter value.

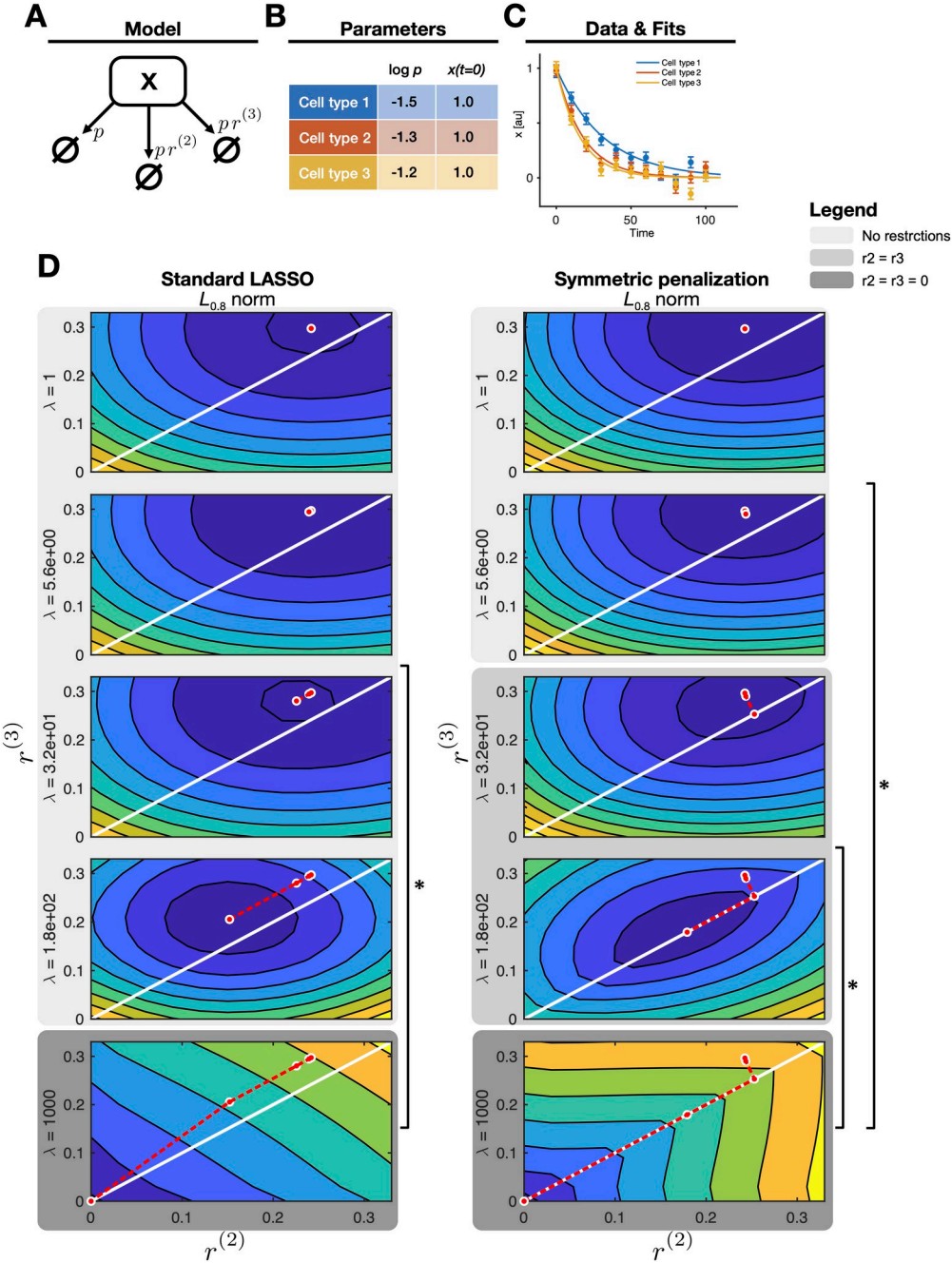

**Fig 2. (A)** Schematic representation of the model structure. Each arrow represents a degradation reaction in one of the three cell types. **(B)** Model parameter values used for simulation, i.e., the kinetic rate constants and initial concentrations used in the three cell types. **(C)** A typical data realization (means and error bars) with un-regularized model fits (lines). **(D)** Objective function landscapes with the regularized best-fit parameter vector (red dot) for different regularization strengths $\lambda$. Square brackets indicate a significant decrease in likelihood in terms of a likelihood ratio test.

**Results.** It can be observed that independent of the regularization approach, for low regularization strengths, the optimization end points are close to but not exactly on the diagonal $r^{(2)} - r^{(3)} = 0$, indicating that the model fit overly adjusts to the specific data realization. Note

that the values of the parameters cannot be directly compared with their true values, since they are biased through the regularization.

For the standard LASSO regularization, the total objective function changes towards a sloped surface with a gradient pointing towards the coordinate origin (Fig 2D, left panel). Consequently, even though starting out closely, optimization end points only reach the diagonal at the origin where $r^{(2)} = r^{(3)} = 0$. This indicates that with this regularization approach, only the outcome of all cell types being equal can be found, which, however, is to be rejected by the likelihood ratio test. Therefore, with standard LASSO regularization, the parsimonious model represents different mutations in cell type 2 and 3.

Regularization with symmetric penalization of fold-changes leads to a different result: Due to the $L_{0.8}$ pseudonorm of the differences between the fold-change parameters, the additional gradient induced by the regularization function implies a curved path towards the diagonal $r^{(2)} - r^{(3)} = 0$ (Fig 2D, right panel). Therefore, in contrast to the standard LASSO regularization, the equal mutation in cell types 2 and 3 is discovered at $\lambda = 32$, before ultimately the optimization end point arrives at the coordinate origin. The likelihood ratio test correctly identifies the former option as the optimal parsimonious model in agreement with the model used to simulate the data.

When employing the $L_1$ norm instead (S1 Fig), the additional gradient induced by the regularization function also implies a path towards the diagonal $r^{(2)} - r^{(3)} = 0$, but it is not curved as for the $L_{0.8}$ pseudonorm and therefore longer. This renders the use of the $L_1$ norm less efficient for identifying common mutations between cell types. In the presented example, the optimization end point reaches the diagonal not until larger regularization strength of $\lambda = 140$.

## Application to a biological model with simulated data

**Model description.** To systematically assess the performance of the symmetric penalization of fold-changes also in a realistic setting, we employ the model of [27] for information processing at the erythropoietin (Epo) receptor, where a mathematical model was established for Epo-receptor dynamics upon ligand binding and calibrated with experimental data, including time-resolved dose-response measurements. This model, which is part of the collection of benchmark models for dynamical modeling of intracellular processes [28], includes six biochemical species, four observables with 85 data points and 16 parameters in total (Fig 3A).

**Approach.** We simulated data in the same configuration as the actual biological data, i.e., the same observables, time points/doses, measurement errors as follows: For the technical reference cell type, we employed the best fit parameters of the original publication. We simulated 100 data sets in three additional cell types (Fig 3B) while varying three of the model parameters, which depicts possible mutations (Fig 3C). We applied regularization with symmetric penalization of fold-changes to all 100 data sets to find common mutations in the five cell types related to the parameters listed in Fig 3C. We fixed the rate constant $k_{ex}$ and the offset parameter to $10^{-5}$ because they were practically non-identifiable with the experimental data set included in Data2Dynamics. Because the parameters of the observation function are typically not related to the investigated biological system, they are excluded from regularization. We then counted how often a fold-change parameter or a difference of fold-change parameters is correctly identified as being compatible with zero in the selected parsimonious model, and repeated the whole process with the standard LASSO regularization for comparison (Fig 3D).

**Results.** For regularization with symmetric penalization of fold-changes, the overall number of correctly identified fold-change parameters and differences of those is very high (Fig 3E). At the same time, there are virtually no false positive results, which is indicated by the absence of cell types identified as having a parameter in common when it is not true (Fig 3C).

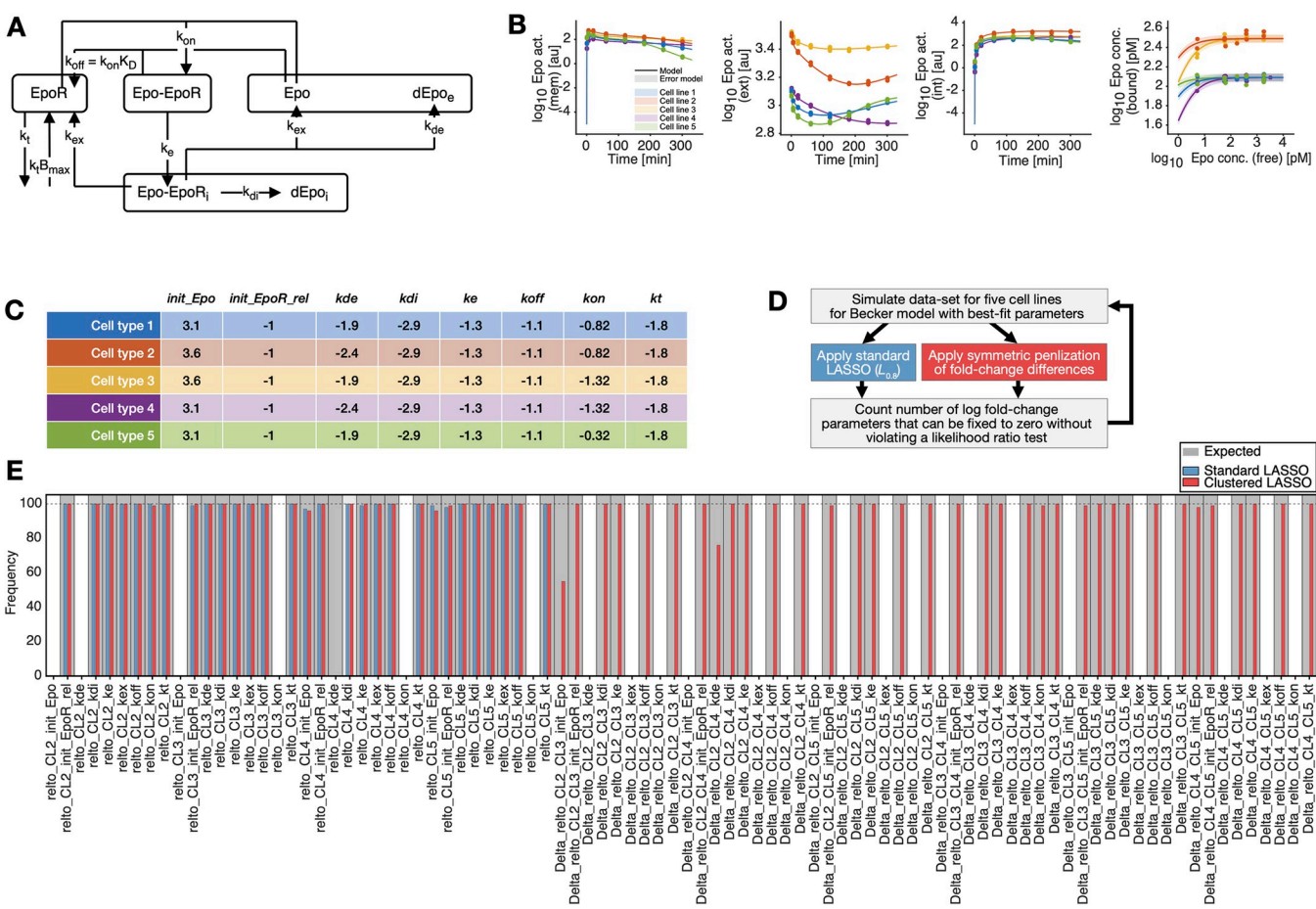

**Fig 3.** **(A)** A schematic representation of the structure of the model used in [27]. **(B)** A typical data realization (dots) with model fits before regularization (lines and shaded areas). **(C)** Parameter values used for simulation of the data for the five cell types. **(D)** Schematic representation of the simulation study workflow. **(E)** The number of times each individual fold-change parameter and difference of fold-change parameters is constrained to zero in the parsimonious model is plotted as a histogram for the standard LASSO approach (blue) and the symmetric penalization of fold-change differences (red). A gray bin background indicates the true values that were used for simulation of the data.

In comparison to the standard LASSO regularization, the proposed method has identical performance w.r.t. the fold-change parameters that relate to the technical reference cell type in this setting. The differences of fold-change parameters equal to zero that represent pairs of cell types with the same mutation are only found using regularization with symmetric penalization of fold-changes. We can observe that in some cases, it is more challenging to identify a fold-change parameter or a difference of them as being zero, e.g., for the difference between the initial value for Epo in cell types 2 and 3. We interpret this as an effect of all parameters being regularized with the same strength $\lambda$, while not all parameters have the same influence on model simulations due to the non-linearity of the models, which however also affects the results of the standard LASSO regularization.

## Application to experimental data

**Model description.** To investigate the effect of ligand addiction, which drives tumor growth, [28] developed a mechanistic model that comprises multiple signal transduction pathways including ErbB, IGF-1R and Met signaling [19]. This model was calibrated on experimental data for seven cancer cell-lines, which was possible through introduction of cell-line

specific parameters for receptor expression, i.e., the initial concentration of the receptor model species. This comprehensive model was later employed to predict proliferation behavior in 58 cancer cell-lines as well as—in combination with decision tree classification—in humans from actual patient data.

**Approach.**   It was reported that it is possible to fit the experimental data of the individual cell-lines with the same model structure assuming equal kinetic rates and cell-line specific receptor abundance based on the standard LASSO approach ([19], Eq 4). Because this method is in general not able to identify subgroups of cell-lines that share the same receptor expression when they do not include the reference cell-line, we investigated the clusters resulting from symmetric penalization of the fold-change parameters. Following the work of [28], we added regularization terms to the objective function for parameters that relate the receptor expression of the cell-lines BxPc3, A431, BT-20, ACHN, ADRr and IGROV-1 to that of H322M. We applied our approach with the original experimental data and 30 regularization strengths ranging between 1 and $10^4$ and identified clusters of cell-lines that share identical receptor expression. We incorporated these clusters into our model and utilized the likelihood ratio test to select the parsimonious model.

**Results.**   Introducing regularization to the optimization function promotes model sparsity, i.e., a low number of cell-line specific parameters (S2A Fig). These constraints result in worse objective function values also in the unregularized setting when compared to the unconstrained model, increasing the likelihood ratio (S2B Fig). We found the optimal value of the regularization strength to be $\lambda = 10^{1.75}$, where the constrained model is still compatible with the full model of [28]. At this value of the regularization strength, we find a number of clusters that share the same receptor expression among the seven cell-lines (S2C Fig). When scanning through different values of $\lambda$, new optima can arise can lead to different combinations of fold-change parameters and differences of such equal to zero. Such behavior can lead to a drop in the test statistic when increasing the value of $\lambda$.

Values for the receptor surface levels for the different cell-lines calculated from data of the CCLE database were reported in [28], which can be compared to the estimated model parameters. We additionally compare their relative amounts to the receptor values from their model that were estimated with penalization of fold-change differences (Fig 4). For the EGF receptor, we find a cluster of the three cell-lines with the lowest EGFR surface level, IGROV-1, ADRr and H322M. For IGF-1R and Met receptors, clustering also resembles the surface receptor value ordering. However, for the ErbB2 and ErbB3 receptors, the clustering of cell-lines resulting from regularized optimization is more challenging to interpret: In ErbB2, clustering seems unrelated to the receptor surface levels, while in ErbB3, cell-lines with similar receptor surface levels are clustered together, but the model estimates for these clusters do not reflect the ordering in the CCLE-based data. However, the latter two effects also occur without regularization in the best-fit reported by [28], which indicates that these results are not related to the symmetric penalization of fold-change differences, but rather a result from the model structure or are artifacts in the CCLE data. Through the identification of clusters of cell-lines that share identical receptor surface levels, we demonstrated how the method we propose provides additional insights compared to the classical LASSO approach.

## V. Conclusions

Regularization is a valuable method to reduce model complexity, e.g., when dealing with multiple cell types. In ordinary differential equation models, it is often applied to infer candidates for parsimonious models from data by clustering similar cell types on the level of individual parameters. Biologically, this can be related to cell types that have identical mutations.

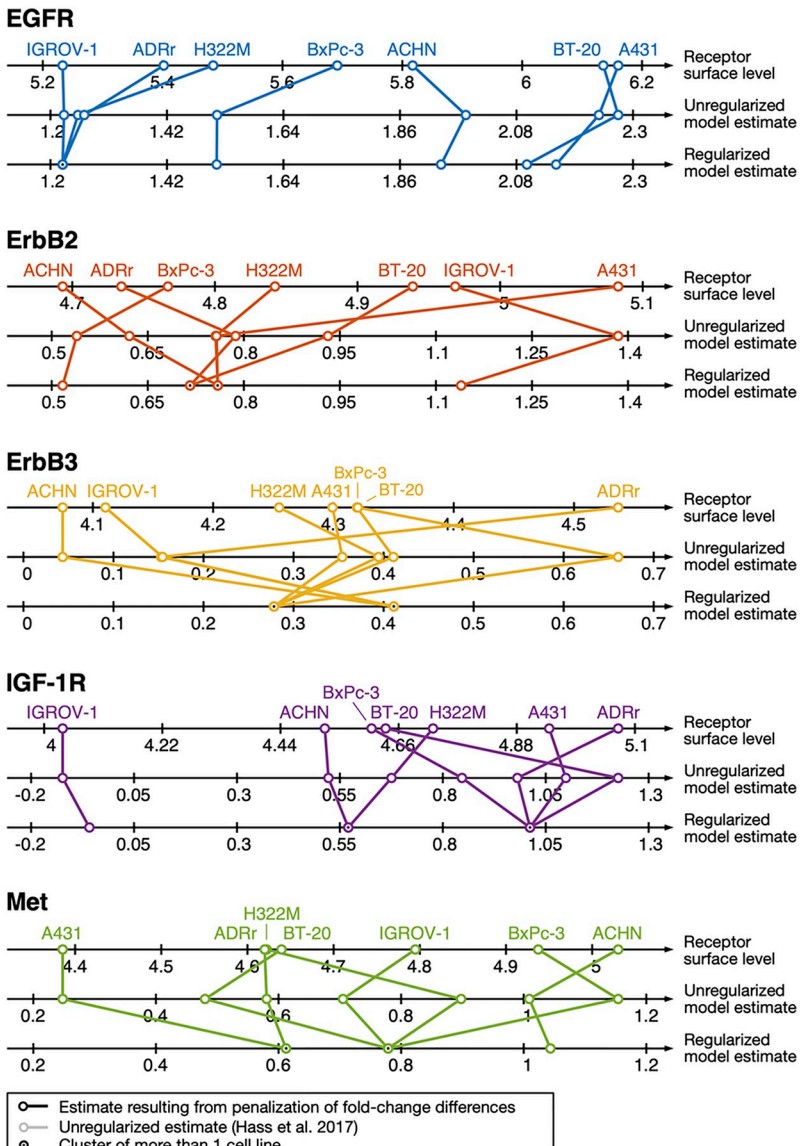

**Fig 4.** Comparison of receptor surface level values from [28]. for EGF, ErbB2, ErbB3, IGF-1 and Met receptors for the different cell-lines H322M, BxPc3, A431, BT-20, ACHN, ADRr and IGROV-1 with the estimated initial values from the mathematical model, as well as the results from regularization with symmetric penalization of fold-change differences.

Approaches based on LASSO regularization with $L_1$ and $L_q$ penalization for two cell types are readily available, but the question of how to handle $n>2$ cell types without biasing the result through the specification of a reference cell type remained open. The grouped LASSO was proposed to address this challenge when predefined groups of parameters can be specified, for example based on prior knowledge. This translates to one parameter being either identical in or specific to all analyzed cell types, without being able to search for subsets that share a certain parameter or mutation.

We proposed an extension of the LASSO approach in differential equation modeling motivated by the clustered LASSO [16] with an $L_{0.8}$ penalization, which is the regularization with symmetric penalization of fold-change differences. We highlighted that this method allows

treating an arbitrary number of cell types without the result being dependent on the arbitrary choice of a reference cell type or introducing any additional parameter. We argued that employing an $L_{0.8}$ instead of the usual $L_1$ norm is more beneficial, because it provides a gradient that leads to more efficient clustering. We discussed how optimization must be adapted in the presence of the new regularization function in terms of the additional residuals and sensitivities. Further, we adapted the optimality criterion and optimizer step truncation accordingly, as well as the calculation of degrees of freedom that is required to perform statistical tests. Concerning computational cost, the computation time required for optimizing a system comprising multiple cell types is multiplied by the number of cell types, which holds true also in a non-regularized setting and is independent from the clustered LASSO approach. A general performance analysis of ODE models can be found in [29].

We demonstrated the advantages of regularization with symmetric penalization of fold-change differences compared to the standard LASSO approach with the $L_{0.8}$ penalization when using more than two cell types in a simplistic example by visualizing the optimization end points of the objective function landscape in parameter space under the influence of increasingly strong regularization. We evaluated the effect on performance of symmetric penalization of fold-change differences in a simulation study: It revealed how our method extends the usefulness of regularization approaches compared to the standard and grouped LASSO, by enabling clustering of any number of cell types that share certain mutations.

We applied our method to a published model of the Epo receptor dynamics upon ligand binding by Becker et al. with realistically simulated data. This confirmed the surplus value of symmetric penalization of fold-change differences compared to standard and grouped LASSO also in a realistic setting. We also revisited the research question of [19] and performed LASSO regularization with symmetric penalization of fold-change parameters on the exact same problem including biological data. We were able to identify clusters of cell-lines that share the same receptor expression, which were confirmed by CCLE-based receptor surface level data where such a comparison is applicable.

We would like to emphasize that, as in all modeling approaches in biology, interpretations of results can be challenging. Moreover, due to data sparsity and limitations of data quality, false positive and false negative results can be obtained, as is also the case for the proposed method. Resulting models should always be seen as a useful approach to understand biological mechanisms and resulting model predictions should subsequently be validated experimentally.

In summary, we illustrated how the proposed method will advance the analysis of multiple cell types. Since we implemented our method in the freely available open-source modeling environment Data2Dynamics, it can be easily applied to a broad range of modeling problems, especially in but not limited to the context of systems biology.

## Supporting information

**S1 Text. Comparison of common values for sensitivities.**
(PDF)

**S1 Fig.** Objective function landscapes with the regularized best-fit parameter vector (red dot) for different regularization strengths $\lambda$ for symmetric penalization of fold-change parameters with the $L_1$ norm. The square bracket indicates a significant decrease in likelihood in terms of a likelihood ratio test.
(PDF)

**S2 Fig.** **(A)** The number of cell type specific parameters dependent on the regularization strength in the model of [28] with symmetric penalization of fold-change differences. **(B)** The likelihood ratio (blue line) between the constrained and unconstrained model increases with the regularization strength. The largest value of $\lambda$ for which the likelihood ratio is below the statistical threshold (red line) represents the parsimonious model (black line). The drop in the test statistic can be accounted to emergence of a new optimum. **(C)** Fold change parameters and differences of them with their values dependent on the regularization strength $\lambda$. Boxes denote the regions where a parameter is not constrained to zero, while the dashed line indicates the regularization strength corresponding to the parsimonious model.
(PDF)

**S3 Fig. Simulation study (Fig 3C) with an ABC model to assess the performance of regularization with symmetric penalization of fold-change differences when using (A) the maximum of sensitivities corresponding to the same residual as a common value, and (B), the mean of sensitivities as a common value.**
(PDF)

## Acknowledgments

We thank Daniel Lill for his suggestions during the conceptualization phase of this study, and Marcel Schilling for his input on the biological relevance of this work.

## Author Contributions

**Conceptualization:** Adrian L. Hauber, Marcus Rosenblatt, Jens Timmer.

**Data curation:** Adrian L. Hauber.

**Formal analysis:** Adrian L. Hauber.

**Funding acquisition:** Jens Timmer.

**Investigation:** Adrian L. Hauber.

**Methodology:** Adrian L. Hauber.

**Project administration:** Adrian L. Hauber.

**Resources:** Jens Timmer.

**Software:** Adrian L. Hauber.

**Supervision:** Marcus Rosenblatt, Jens Timmer.

**Validation:** Adrian L. Hauber.

**Visualization:** Adrian L. Hauber.

**Writing – original draft:** Adrian L. Hauber, Marcus Rosenblatt.

**Writing – review & editing:** Adrian L. Hauber, Marcus Rosenblatt, Jens Timmer.

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
