## [Decision Letter · Decision Letter 0]

3 Apr 2023

Dear Mr. Hauber,

Thank you very much for submitting your manuscript "Uncovering specific mechanisms across cell types in dynamical models" for consideration at PLOS Computational Biology.

As with all papers reviewed by the journal, your manuscript was reviewed by members of the editorial board and by several independent reviewers. In light of the reviews (below this email), we would like to invite the resubmission of a significantly-revised version that takes into account the reviewers' comments.

We cannot make any decision about publication until we have seen the revised manuscript and your response to the reviewers' comments. Your revised manuscript is also likely to be sent to reviewers for further evaluation.

Sincerely,

Kiran Patil

Section Editor

PLOS Computational Biology

Reviewer's Responses to Questions

**Comments to the Authors:**

Reviewer #1: See attached file

Reviewer #2: The authors of this paper suggest utilizing clustered LASSO regularization in dynamical models described by ordinary differential equations. The motivation for such regularization is to enable the identification of mechanisms specific to certain cell types, whatever the number of cell types, without having a bias introduced by selecting a particular reference cell type. It seems it is a follow-up of previous research conducted by one of the co-authors, as mentioned in references [12] and [15]. To showcase the proposed regularization, the authors present three case studies.

The text is well-structured, technically correct, and relatively clear. Below are some remarks that might improve the manuscript.

Comments:

1. From Figure 1b, it is clear that when Lambda is sufficiently large, the parameter estimation process will strongly favor maximizing the number of parameters with identical values due to symmetric penalization. Although this is the main reason for implementing this form of regularization, it can lead to the identification of multiple false positives, such as several mutations that are identified to be common but are not. This is especially probable in underdetermined problems, where parameters can vary widely due to the scarcity of observations, as noted by Gutenkunst et al (PloS Comp Bio, 2007). Since the authors focus on biological systems as examples, which generally lack sufficient observations compared to the number of parameters that need to be identified, this issue should be explicitly mentioned and discussed. Can we avoid this problem at all?

2. Related to my previous remark, in the presented toy example, it is possible for cell types 2 and 3 to have distinct mutations, yet the proposed algorithm might identify them as common. Moreover, since clustered LASSO does not yield the actual parameter values (instead, it tends to shrink them toward zero, like standard LASSO), it is unclear why one should apply this regularization in such a scenario. In general, the authors should provide clear guidance to readers on when it is appropriate and when it is not advisable to use their regularization method.

3. Given that linear correlations between parameters are typical in various systems, I would like to know if this regularization method can be extended to address such cases. I assume that the correlation coefficients would probably appear as the optimization variables in the penalty term. Can you comment on this?

4. The explanation in lines 310-320 and Figure 2 could be clearer regarding the toy model. It is hard to distinguish what the authors consider as reality and what is meant to provide “realistic” data for the example. Moreover, in the text, the species is denoted with x, in Figure as A. The value for parameter p in the text is logp=-1.5 whereas in Figure 2b we find -2 (I was trying to deduce based on the graph in Figure 2c what was used). Next, the values in Figure 2b are not p (as written at the top of the table), these should be log p. Similarly in Figure 2A, you have a mix of notations. With the notation presented earlier in the manuscript, I would not say that you can write p1r1^(2) and p1r1(3).

5. Regarding the size of systems this kind of regularization can be used on, how does this method scale computationally when the number of cell types increases and the number of parameters? Please provide the computational time estimates.

6. Since this regularization is proposed in the context of nonlinear optimization, how can you guarantee you found a satisfying result? Furthermore, can you enumerate all alternative solutions of clustered parameters? For example, in the study depicted in Figure 4, there are cell line groups that have the same parameter values. Will these groups change if you converge on a different solution?

7. Lines 240-250, to avoid that r_i^(j)-r_i^(k)=0, r_i^(j) <>0 occurs, would it work using a kind of forgetting factor?

8. The notation throughout the paper is confusing. For example, lines 120-126, one understands that in r_i^(j) the subscript i denotes the i-th parameters, whereas the superscript j denotes the cell type. But then, lines 199-200, we see the same subscript i denoting “the l-th and the o-th parameter”. Furthermore, lines 210-212, what the subscript l-n denotes. In general, for most superscripts and subscripts, their range is missing.

9. Line 347, there is a floating sentence fragment.

Reviewer #3: Hauber, Rosenblatt and Timmer present the development of LASSO regularization in context of estimating parameters of dynamic models of cellular pathways. Their method is a generalization of the LASSO regularization technique for n > 2 conditions, which enables reference-free distinction of samples, based on their estimated parameter values. Instead of penalizing changes in parameters of a perturbed cell state, compared to a predefined reference parameter set, their grouped LASSO approach penalizes parameter differences across all possible pairs of samples. Under the assumption that all conditions can be distinguished only by changes in parameter values with respect to each other, and not by structural model changes, their method provides a minimal parameter change to describe different conditions. Furthermore, the authors propose a modification to MATLAB’s lsqnonlin optimizer to account for the additional penalty terms. In general, this approach is relevant for improving cell state distinction through dynamic modelling, as it reduces bias based on experimental group selection, and enables robust identification of multiple subgroups within predefined experimental groups.

The authors provide three examples, illustrating the application of their proposed regularization approach, ranging from fully simulated to completely experimental, where an experiment by Hass et al. (2017) is repeated, incorporating the proposed grouped LASSO approach. A major concern, however, relates to the method of hyperparameter tuning. In the proposed approach, selection of the hyperparameter occurs by performing a likelihood ratio test on the fitted likelihood values. While the approach taken seems justified in general, the likelihood ratio test should instead be performed on likelihood values calculated on samples unseen during the parameter estimation procedure. This prevents biasing the results towards the specific samples included in the data. This reviewer recommends including a cross-validation approach to strengthen the hyperparameter selection.

Additionally, the following minor points should be addressed and possibly included:

1) In the explanation on parameter estimation in dynamical systems, the authors mention that parameters are inferred from normally distributed data. It seems that the authors may have meant normally distributed error in the data, as the measurements that are used to fit the dynamic model are not at all normally distributed.

2) In the section Adaptations to the optimization algorithm, two parameters are introduced with subscripts l and o in the first subsection, while in the subsequent subsections, these parameters seem to be referred to using subscripts l and n. Please improve the consistency in the notation.

3) In the formulae for the residual and the sensitivity of the residual in the same section as in the previous comments, the norm is divided by 1/lambda, which could be made more clear by multiplying the norm with lambda instead.

**Have the authors made all data and (if applicable) computational code underlying the findings in their manuscript fully available?**

Reviewer #1: Yes

Reviewer #2: Yes

Reviewer #3: Yes

PLOS authors have the option to publish the peer review history of their article (what does this mean?). If published, this will include your full peer review and any attached files.

Reviewer #1: No

Reviewer #2: No

Reviewer #3: No
---

## [Decision Letter · Decision Letter 1]

14 Aug 2023

Dear Dr. Hauber,

We are pleased to inform you that your manuscript 'Uncovering specific mechanisms across cell types in dynamical models' has been provisionally accepted for publication in PLOS Computational Biology.

Best regards,

Kiran R. Patil, Ph.D.

Section Editor

PLOS Computational Biology

Reviewer's Responses to Questions

**Comments to the Authors:**

Reviewer #4: This reviewer would like to initially thank the authors for taking his comments into account in this revised version. The revised version of the authors’ submission ‘Uncovering Specific Mechanisms across Cell Types in Dynamical Models’ improved upon the initial submission in readability and clarity of the presented work. In addition to the updated components, recommended in my previous review, I would like to explicitly express my appreciation for the addition on interpretation of results in biological models. In this submission, the authors present their method not as being able to solve these major challenges in biological modelling, but as a method that reduces bias in modelling (possibly unknown) subgroups in the data and improves the robustness of parameter estimates against small differences in measurement data between these groups. Concerning the submission overall, I have only one small remark:

1) The numbering in the references section is missing numbers 25 and 26 (while 25 is present in the version of the manuscript indicating the changes that were made). This might be a result of the addition and removal of references during revision.

**Have the authors made all data and (if applicable) computational code underlying the findings in their manuscript fully available?**

Reviewer #4: Yes

PLOS authors have the option to publish the peer review history of their article (what does this mean?). If published, this will include your full peer review and any attached files.

Reviewer #4: No

---

## [Editor Report · Acceptance letter]

8 Sep 2023

PCOMPBIOL-D-23-00034R1 

Uncovering specific mechanisms across cell types in dynamical models

Dear Dr Hauber,

I am pleased to inform you that your manuscript has been formally accepted for publication in PLOS Computational Biology. Your manuscript is now with our production department and you will be notified of the publication date in due course.

With kind regards,

Zsofia Freund
